# Association of Paraoxonase-1 and NT-proBNP with Clinical, Clinico-Pathologic and Echocardiographic Variables in Dogs with Mitral Valve Disease

**DOI:** 10.3390/vetsci10010033

**Published:** 2023-01-01

**Authors:** Diana Rammal, Christos K. Koutinas, Labrini V. Athanasiou, Melpomeni Tangalidi, Camila P. Rubio, José J. Cerón, Androniki Tamvakis, Michael N. Patsikas, Zoe S. Polizopoulou

**Affiliations:** 1Diagnostic Laboratory, School of Veterinary Medicine, Aristotle University of Thessaloniki, 54627 Thessaloniki, Greece; 2Companion Animal Clinic, School of Veterinary Medicine, Aristotle University of Thessaloniki, 54627 Thessaloniki, Greece; 3Clinic of Medicine, Faculty of Veterinary Medicine, University of Thessaly, 43131 Karditsa, Greece; 4Interdisciplinary Laboratory of Clinical Analysis of the University of Murcia, Regional Campus of International Excellence ‘Campus Mare Nostrum’, University of Murcia, Campus de Espinardo s/n, Espinardo, 30100 Murcia, Spain; 5Laboratory of Ecology and System Dynamics, Department of Marine Sciences, University of the Aegean, 81100 Mytilene, Greece; 6Laboratory of Diagnostic Imaging, School of Veterinary Medicine, Aristotle University of Thessaloniki, 54627 Thessaloniki, Greece

**Keywords:** mitral valve disease, NT-proBNP, Paraoxonase-1, dog

## Abstract

**Simple Summary:**

In order to identify the association of Paraoxonase-1 and N-terminal-prohormone-B-type natriuretic peptide with selected clinico-pathologic and echocardiographic parameters in mitral valve disease, 80 dogs in various clinical stages were enrolled. Paraoxonase-1 concentrations were not correlated with clinical stage, gender or concurrent conditions. At the same time, it was inconsistent in showing significant changes against distinctive echocardiographic and clinico-pathologic parameters. N-terminal-prohormone-B-type natriuretic peptide concentration was expectedly correlated with clinical stage and echocardiographic indices of cardiomegaly and heart failure, but not with Paraoxonase-1 activity. These findings suggest that Paraoxonase-1, compared to N-terminal-prohormone-B-type natriuretic peptide, is an insensitive marker for the severity of mitral valve disease and that its utility may be hampered by confounding factors.

**Abstract:**

The objective of the present study was to measure the concentration of Paraoxonase-1 (PON-1) and N-terminal-prohormone-B-type natriuretic peptide (NT-proBNP), in the serum of dogs with degenerative Mitral Valve Disease (MVD), in order to identify their association with the clinical stage and specific clinico-pathologic and echocardiographic findings.Eighty dogs diagnosed with MVD and staged according to the ACVIM (American College of Veterinary Internal Medicine) consensus statement (B1, B2, C and D), based on their clinical, radiographic, and echocardiographic findings, were included in the study. NT-proBNP was measured only in stage B1 and B2 dogs. Clinical stage did not have a significant effect on PON-1 concentrations (*p* = 0.149), but NT-proBNP levels were lower in B1 dogs (*p* = 0.001). A significant correlation between PON-1 and total plasma proteins (*p* = 0.001), albumin (*p* = 0.003) and white blood cell count (*p* = 0.041) was detected, whereas there was no significant correlation (*p* = 0.847) between PON-1 and NT-proBNP concentrations. PON-1 showed a significant but weak negative correlation with normalized left ventricular internal diameter at diastole (LVIDdn) (*p* = 0.022) and systole (LVIDsn) (*p* = 0.012), as well as mitral valve E to A wave velocity ratio (MV E/A) (*p* = 0.015), but not with Left Atrial to Aortic root ratio (LA/Ao) (*p* = 0.892) or fractional shortening (FS%) (*p* = 0.944). PON-1 seems to be an insensitive marker of clinical stage and disease severity in MVD, but can be indicative of some clinico-pathological and echocardiographic changes. NT-proBNP changes are independent of oxidative stress.

## 1. Introduction

Chronic mitral valve disease (MVD) is the most common heart disease in middle-aged to elderly dogs, as well as the most common cause of congestive heart failure in this species [1,2,3]. The incidence of the disease in the dog is age-, breed-, gender-, and body size-related [4,5,6,7]. Typically, the disease in small breed dogs progresses gradually over a period of several years, from mild to advanced congestive heart failure, even though many dogs remain in the asymptomatic stage [8].

Recent studies have shown a direct relationship between the disease and oxidative stress. More specifically, they have shown that mitral valve disease appears to be related to increased concentrations of both acute phase proteins (e.g., C-reactive protein, A-amyloid, ceruloplasmin, ferritin) and oxidative stress markers (e.g.,paraoxonases, butyrylcholinesterase, vitamin E) [9,10,11,12]. Oxidative stress is an imbalance between the production of active oxygen species and the antioxidant defense mechanisms of an organism and plays an important role in the etiology of certain metabolic diseases [13]. There has been some interest in the role of paraoxonases in the pathogenesis of cardiovascular diseases in humans [14,15,16,17,18]. These proteins appear to perform multiple functions in various biochemical pathways, such as protecting against oxidative damage and lipid peroxidation, enhancing natural immunity, detoxifying the organism from active molecules, and regulating cell proliferation and cellular apoptosis. They are capable of performing multiple autonomous and independent functions. Paraoxonase-1 (PON-1) is synthesized in the liver and is mainly associated with high density lipoproteins (HDL).It is considered an indicator of inflammation and oxidative damage, with its concentration decreasing as oxidative stress progresses (negative biomarker) [19]. PON-1 also seems to protect from vascular disease and is used as a biomarker [20] in diseases related to three conditions: (a) oxidative stress, since it protects against peroxidation [14]; (b) inflammation, since it is considered a negative acute-phase protein [21]; (c) hepatic diseases, because it is synthesized in the liver [22].

Natriuretic peptides are released by the cardiac muscle in response to various stimuli [23,24], with the most common type of peptides for canine and feline species being the B-type natriuretic peptide (BNP) [25]. The inactive proBNP form has the advantage of being more stable than the active BNP form, which allows it to be used as a diagnostic tool in cardiac diseases [26]. BNP is released by the heart in response to cardiac stress or injury such as hypertrophy, volume overload, and hypoxia [27]. Cardiomyocytes thus respond to cardiac dilation by increasing N-terminal-prohormone-B-type natriuretic peptide (NT-proBNP) production, which is increasingly used for the prognosis of MVD progression. Several studies showed that NT-proBNP levels increased with the progression of MVD, relying on quantification of mitral regurgitation volume, left atrial size, and left ventricular diameter as well as other echocardiographic variables, as a major determinant for disease progression. Asymptomatic dogs with higher levels of serum NT-proBNP were at a higher risk of disease progression [28,29,30].

The aim of this study was to measure the concentration of PON-1 and NT-proBNP in the serum of dogs with MVD, in order to investigate their association with the clinical stage as well as specific clinico-pathologic and echocardiographic findings.

## 2. Materials and Methods

Dogs included in the study were admitted for cardiologic consultation at the Companion Animal Clinic of the Faculty of Veterinary Medicine, Aristotle University of Thessaloniki, either as novel cases or re-examinations. Upon admission, all dogs underwent a thorough physical and cardiological examination (electrocardiography, blood-pressure measurement, echocardiography), performed by the same examiner (CKK). The presence of symptoms of related congestive heart failure (cough, syncope, exercise intolerance, dyspnea, tachypnea, cyanosis, and ascites) as well as the intensity of the cardiac murmur and its point of maximal intensity were recorded. At the same time, gender, age, reproductive status, and any concurrent conditions of major or minor clinical significance were documented. Based on the clinical and cardiologic examination, the clinical stage was determined for each dog (B1, B2, C and D according to the American College of Veterinary Internal Medicine classification scheme).

In order for the examined animals to be enrolled in the study, the following inclusion criteria had to be fulfilled:Diagnosis of MVD, made by detection of an auscultable regurgitant murmur in asymptomatic dogs or those with heart failure, and confirmedby 2D and Doppler echocardiography (thickened, nodular or prolapsing mitral valve leaflets, and presence of regurgitant blood volume). Concurrent tricuspid valve degeneration was not a criterion for exclusion since the disease progress is the same.No evidence of clinical signs and/or clinico-pathological abnormalities indicative of a systemic or severe disease that may affect oxidative stress, such as kidney disease, liver disease, or pronounced diffuse/localized inflammation and endocrinopathies. This evidence was based on signalment, clinical examination, and basic hematology and biochemistry profiles, as well as radiographic evaluation.

This study protocol was approved by the Research and Ethics Committee and the General Assembly of the Faculty of Veterinary Medicine, Aristotle University of Thessaloniki (ref. no. 69/30-06-2016). The owners of animals participating in the study signed an informed consent form.

Blood samples (5 mL) were collected from all dogs for complete blood counts (CBC) (whole blood), *Dirofilaria immitis* antigen detection, and routine biochemistry (serum). The animals were fasted for 12 h before specimen collection, which was conducted between 10.00 a.m. to 14.00 p.m. in all cases. One mL of blood was immediately transferred into tubes containing K-EDTA (Vet Collect tubes, Idexx, Wetherby, UK). Plasma for the NT-proBNP measurement was harvested from the K-EDTA samples after CBC evaluation and transferred into plain plastic tubes (Eppendorf microtubes, Eppendorf AG, Hamburg, Germany). From the remaining 4 mL of serum, one aliquot was used for routine biochemistry (Eppendorf microtubes, Eppendorf AG, Hamburg, Germany) and two aliquots were saved in cryotubes (Provett per Criogenia, Biosigma, Cona, Italy) to measure PON-1 concentration. The specimens for the common laboratory measurements were processed immediately, whereas those reserved for biomarker (PON-1 and NT-proBNP) measurement were immediately stored at −80 °C until shipment, which occurred within approximately 3 months for each sample batch. Sample batches were shipped frozen in special containers with dry ice and were thawed only once, just prior to analysis.

For CBC and routine serum biochemistry, an automated hematology (ADVIA 120 Hematology System, Bayer Diagnostics, Dublin, Ireland) and a clinical chemistry analyzer (Flexor E, Vital Scientific, Spankeren, The Netherlands) were used, respectively.

A complete cardiologic evaluation included the measurement of systolic blood pressure (SBP) using a high-definition oscillometric device (Vet-HDO, Systeme + Beratung medVET Gmbh, Babenhausen, Germany), electrocardiography (Cardico 306 ECG, Kenz, Japan), and a full echocardiographic examination (GE Vivid-I, Veterinary Ultrasound, Chicago, IL, USA). When necessary, left lateral and dorsoventral thoracic radiographs were performed to assess cardiomegaly and pulmonary congestion.

The echocardiographic parameters studied included the ratio of left atrium to aortic root diameter (LA/Ao), which was obtained from the right parasternal 2D-view at the base of the heart, and the M-mode left ventricular diameter during both diastole and systole, which was obtained at the level of the papillary muscles from the right parasternal short-axis view. This was normalized for body weight using Cornell’s allometric scaling (LVIDdn, LVIDsn) [31]. Fractional shortening (FS%) was automatically calculated based on the measurements outlined above. Mitral valve E wave peak velocity (MV E) and E to A wave ratio (MV E/A) was measured using Doppler echocardiography from the left parasternal apical 4-chamber view.

### 2.1. Methodology for Measuring Paraoxonase-1 (PON-1)

PON-1 activity was measured using a previously validated assay, with p-nitrophenyl acetate as substrate. The process was adapted for an automated analyzer [32].

### 2.2. Methodology for Measuring NT-proBNP

NT-proBNP was measured with a second-generation sandwich ELISA assay, as previously described and validated [33].

### 2.3. Statistical Analysis

Both the target variables of PON-1 and NT-proBNP were tested for the normality assumption using either the Shapiro–Wilk test or the corresponding Q–Q scatterplot. Moreover, the homogeneity of variance was assessed using the Levene test.

The effect of the clinical stage on the concentration of PON-1 was tested with ANOVA statistical analysis, while the effect of binary factors (including gender and the occurrence of coexisting conditions) on PON-1 was determined using the *t*-test for two independent samples. On the other hand, since the NT-proBNP variable did not follow normal distribution, the corresponding non-parametric tests were used. Thus, the effect of the clinical stage and gender on the concentration of NT-proBNP was determined using the Mann–Whitney test (Wilcoxon Rank Sum). Finally, the correlation of PON-1 and NT-proBNP with various laboratory and echocardiographic parameters was defined using Spearman’s rank coefficient testing. The level of significance for all tests was set to a *p* = 0.05.

## 3. Results

A total of 80 dogs with degenerative mitral valve disease were included in the study and divided into four groups based on their initial clinical presentation (B1, B2, C and D). Thirty-one dogs (38.75%) were allocated in group B1 (asymptomatic, without cardiomegaly), thirty dogs (37.5%) in group B2 (asymptomatic, with cardiomegaly, enrolled prior to receiving any medication), twelve (15%) in group C (symptoms of heart failure, receiving a combination of furosemide, spironolactone, benazepril and pimobendan, in doses tailored for clinical effect), and seven (8.75%) in group D (dogs with advanced heart failure and not responding to the standard treatment for stage C described above).

The age of the dogs ranged from four to 16 years (median 11 years) and their body weight from 2.1 to 25 kg (median 7.75 kg). Forty-eight out of 80 (60%) were male and 32 (40%) were female. Of the female dogs, 30 were neutered. Fifty-nine dogs (73.75%) belonged to pure breeds, the most common of which were Cavalier King Charles Spaniel (14/80), Miniature Pinscher (9/80), Miniature Poodle (6/80), Yorkshire Terrier (5/80), Pekingese (5/80), and Maltese (4/80), whereas 21/80 were of mixed breed.

Symptoms of dogs upon clinical presentation are represented as follows. Presenting complaints of dogs included cough (20/80, 25%), syncope (5/80, 6.25%), exercise intolerance (6/80, 7.5%), dyspnea (9/80, 11.25%), tachypnea (29/80, 36.25%), cyanosis (1/80, 1.25%), weak pulse (1/80, 1.25%), and ascites (6/80, 7.5%).

The most common concurrent condition documented was mild-to-moderate periodontal disease, seen in three of B1, six of B2, five of C, and two of stage D dogs. Other conditions identified included idiopathic epilepsy (one dog of stage B1), chronic bronchial collapse (a single B2 dog), and idiopathic vestibular syndrome and single mammary nodule (one dog of group D, respectively). Concurrent tricuspid valve degeneration that resulted in moderate-to-severe regurgitant volume, was evident in 8/80 (10%) dogs.

Basic descriptive statistics of various clinical, laboratory, and echocardiographic parameters within each clinical stage of MVD are presented in Table 1 (first columns). Age, heart rate, and creatinine concentrations increased along with the clinical stage while ALB showed a reverse tendency. Most of the echocardiographic variables (i.e.,LVIDdn, LVIDsn, LA/Ao, FS%, MV E/A, MV E) increased from B1 through B2 to stage C but seemed to decrease in stage D dogs.

The correlation analysis among the various echocardiographic and clinico-pathological parameters (Table 1, last two columns) showed asignificant positive correlation of PON-1 activity with the total plasma proteins (r = 0.480, *p* < 0.001) and albumin concentration (r = 0.431, *p* < 0.001). On the other hand, PON-1 was found to be negatively associated with the white blood cell count (r = −0.242, *p* < 0.05), but showed no significant correlation with the systolic blood pressure nor with the dog’s age (*p* > 0.05). Regarding echocardiographic parameters, PON-1 was only moderately negatively correlated with MV E/A ratio, LVIDdn and LVIDsn (r = −0.277, −0.257 and −0.283 with *p* < 0.05, respectively), but not with LA/Ao, MV E or FS (*p* > 0.05). The association between PON-1 activity and NT-proBNP concentration was not found to be significant (r = −0.028, *p* = 0.847). As for the NT-proBNP concentration, it was found to be significantly correlated with LA/Ao (r = 0.582, *p* < 0.001) and LVIDdn (r = 0.324, *p* < 0.05). Finally, NT-proBNP had a low positive correlation with the animal’s age (r = 0.384, *p* < 0.01).

The activity of PON-1 ranged from 2.52 to 6.53 IU/mL in the dogs included in our study. Furthermore, PON-1 activity in dogs with MVD (*n* = 80) was analyzed to determine differences among the dogs’ various clinical stages (Table 2). PON-1 activity did not show a stable tendency across the MVD stages, as it recorded the highest values in stage C and the lowest in stage D. The statistical analysis showed that the clinical stage in dogs with MVD was not associated with PON-1 activity (ANOVA, *p* > 0.05). Moreover, PON-1activity showed no significant differences between females and males (*t*-test for independent samples, *p* = 0.619), nor between dogs with and without coexisting conditions (*t*-test for independent samples, *p* > 0.05).

NT-proBNP concentration was only measured in 50/80 dogs belonging to groups B1 (27/50) and B2 (23/50) and ranged from 250 to 3417 pmol/L. The basic statistical data within the examined groups are presented in Table 3. In contrast to PON-1, NT-proBNP showed considerably greater values when measured in dogs of clinical stage B2 than those of B1. This finding was also confirmed by the statistical analysis, which showed significant differences in NT-proBNP concentration between the two clinical stages (Mann–Whitney U test, *p* < 0.001). NT-proBNP concentration was also significantly higher in dogs with coexisting conditions (Mann–Whitney U test, *p* < 0.05). Finally, the gender of the dogs was not found to be significantly correlated with NT-proBNP concentration (Mann–Whitney U test, *p* > 0.05).

## 4. Discussion

Previous studies have demonstrated the value of inflammation biomarkers in identifying the inflammatory process in the various stages of heart failure [34,35,36]. Even though these biomarkers are not tissue-specific, and their concentrations change in various heart diseases [37,38], the fact that these changes are associated with the progression of heart failure may indicate at least the partial contribution of inflammation in the pathogenesis of mitral valve disease and heart failure [34,37]. However, thus far the inflammation biomarkers have been unable to differentiate between the asymptomatic and mildly symptomatic clinical stages of heart failure, increasing or decreasing significantly only during advanced stages [34,35]. In contrast to specific cardiac biomarkers (cTnI and NT-proBNP), which appear to have significant correlation with the stage of heart failure in degenerative valvular disease [10,39], PON-1 does nοt appear to share validity. Non-specific changes in PON-1 activity can be observed by the systemic inflammation caused by advanced heart failure, and possibly also by liver congestion in stage C and D dogs [40]. In humans, the involvement of oxidative stress in the early stages of mitral valve degeneration has been described through the measurement of PON-1 [41]. The minor differences between the subclinical stages of the dogs in our study indicate that if oxidative stress eventually plays an important role in the progression of heart failure, PON-1 activity does not appear to be a reliable marker. In other studies [12], stage D dogs had lower PON-1 activity when compared to stages A and B, but both MVD and dilated cardiomyopathy patients were included. Specifically, in the same study, most stage D dogs belonged to the dilated cardiomyopathy group. Conversely, our underlying cardiac disease population was more uniform. At the same time, PON-1 activity seems to increase as soon as treatment is instituted, especially in symptomatic dogs [12]. All our stage D dogs were treated with multiple cardiovascular medications, which would explain the non-significant reduction in PON-1 activity in that particular stage. Another reason for the lack of difference between clinical stage groups could be the imbalance between the numbers of asymptomatic dogs (B1 and B2) compared to those with advanced heart failure (stage D) in our study (61 versus seven). Finally, our results seem to agree with a similar study utilizing other oxidative stress markers, such as oxidized low-density lipoprotein [42].

On the other hand, lowering of PON-1 activity was associated with an increase in LVIDdn and LVIDsn measurements, as well as MV E/A ratio, which are markers of advanced cardiac disease and decompensated heart failure [43,44], regardless of group. In a recent study, similar conclusions were reached regarding the correlation of PON-1 and left ventricular dimensions [12]. This result suggests that echocardiographic parameters might be a more suitable indicator of systemic inflammation in heart failure, compared to the traditional staging scheme. However, it has also been reported that PON-1 activity may actually increase as clinical stage progresses [45]. At the same time, the fact that LA/Ao, MV E, and FS% were not similarly correlated to changes in PON-1 activity makes this explanation rather implausible. The lack of correlation between LA/Ao, MV E, and PON-1 in our study, despite the two parameters also being markers of cardiomegaly and congestion, respectively, is difficult to explain. At the same time, however, the low coefficient of both left ventricular diameters (systolic and diastolic) suggests that the correlation between cardiomegaly and PON-1 is weak at best. The causal relationship between cardiac remodeling and oxidative stress has not been elucidated, although it has been described in humans [46].

The activity of PON-1 has been found to be significantly reduced in dogs with leishmaniosis, ehrlichiosis, acute pancreatitis, parvovirus enteritis, and endotoxemia [47,48,49,50,51]. We were very thorough in excluding dogs with systemic diseases, even though MVD patients are usually affected by other comorbidities [52,53,54]. In our study, mild-to-moderate periodontal disease as well as other traditionally non-inflammatory conditions were observed in some cases. However, they did not seem to affect PON-1 activity. In the case of periodontal disease, oxidative stress, when measured by the concentration of nitric acid products, did not appear to change between dogs with different degrees of periodontal inflammation prior to dental procedures [55]. It is unlikely that this could have impacted our results, since there was no difference in PON-1 activity between the stages of heart failure, and periodontal disease was represented in each group.

The role of gender in assessing oxidative stress has been described in previous studies in rats, dogs, and humans, with females generally having lower concentrations of PON-1 [41,42,56]. However, in our study, the results did not change significantly between genders. This discrepancy could be due to the large number of spayed females (30/32, 93.75%) included in the study.

Lipemia, as defined by an increase in the concentration of triglycerides and cholesterol, as well as prolonged freezing (−20 °C for over 6 months), appears to increase PON-1 activity when measured by the automated method based on paraoxone [57,58,59]. At the same time, changes in production of high-density lipoproteins due to inflammation may affect the production and activity of PON-1 [16,59]. In our case, dogs were fasted for at least 8 h prior to blood collection and blood samples were transferred to deep freezing storage (−80 °C).

PON-1 activity was associated positively with albumin concentration, and negatively with total protein and WBC. This is since the total proteins are the sum of circulating albumin and globulin, and albumin reduction is an indicator of chronic inflammation and of oxidative stress that accompanies the inflammatory process. The positive association between albumin and PON-1 has been described in dogs and humans in various diseases such as parvovirus enteritis and nephrotic syndrome [50,60,61]. The correlation of heart failure and inflammation appears to be supported by the decrease in albumin concentration and the rise in other inflammatory biomarkers, such as C-reactive protein (CRP) [35]. However, PON-1 appears to have no direct correlation with CRP, except in those cases where the concentration of the latter is particularly high [57].

NT-proBNP concentrations were higher in stage B2 dogs in comparison to B1 dogs, as has been described in numerous other studies [24,62]. At the same time, LA/Ao and LVIDdn were strongly correlated with this cardiac biomarker. While this result was to be expected, there was no correlation, however, with PON-1 activity, which has not been studied so far according to our knowledge. This reveals that volume overload, as indicated by NT-proBNP, might not be associated with oxidative stress.

The limitations of this study include the absence of age-matched healthy controls, and the cross-sectional design of the study. At the same time, the necessity for treatment customization in stages C and D of MVD creates an unstable variable that may have affected PON-1 measurements in these dogs. The same holds true regarding disease duration. However, the aim of the study was to assess oxidative stress, as indicated by PON-1 activity, between heart failure stages. In other studies, controls seemed to have either lower PON-1 [45] or similar PON-1 activity [12] compared to the asymptomatic stages of MVD, making oxidative stress markers rather insensitive in differentiating between normal and early MVD dogs. In one of those studies, the measurements in the control group were made by the same laboratory, using the same method as ours, and providing a median/interquartile range of 3.9/3.5–4.6 IU/L [12]. Although this is a cross-sectional study, it is part of a larger and currently ongoing PhD protocol (DR) that aims to study the longitudinal concurrent changes of PON-1 and NT-proBNP in asymptomatic dogs. Regarding lipoprotein measurement, it would have been ideal to have a full lipid profile in these dogs (HDL, LDL, cholesterol, triglycerides), which could have allowed for a standardized PON-1 measurement as suggested by [45].

## 5. Conclusions

Oxidative stress, as shown by the decrease in PON-1 activity in dogs with degenerative mitral valve disease, is not correlated with clinical staging or with NT-proBNP concentrations.

PON-1 activity has a weak negative correlation with left ventricular dimensions, as measured by echocardiography, even though it is independent of left atrial size or indices of systolic function.

NT-proBNP concentration is significantly higher in stage B2 compared to B1 and is significantly and strongly correlated with echocardiographic parameters of cardiomegaly and congestion.

## Figures and Tables

**Table 1 vetsci-10-00033-t001:** Basic descriptive statistics (in terms of mean and interquartile range) within the clinical stage of MVD and correlation analysis with PON-1 and NT-proBNP of various clinical, clinico-pathological, and echocardiographic parameters. (MVD: Mitral Valve Disease; PON-1: Paraoxonase-1; NT-proBNP: N-terminal-prohormone-B-type natriuretic peptide; N/A: not applicable; TP: Total proteins; ALB: Albumin; CREA: Creatinine; LVIDdn: Left ventricular internal diameter at diastole normalized;LVIDsn: Left ventricular internal diameter at systole normalized; LA/Ao: Left atrial to aortic root ratio; FS%: Fractional shortening; MV E: Mitral valve E wave velocity; MV E/A ratio: Mitral valve E wave to A wave velocity ratio).

Parameters	Clinical Stage of MVD				Correlation Coefficient	
	B1 (*n* = 31)	B2 (*n* = 30)	C (*n* = 12)	D (*n* = 7)	PON-1	NT-proBNP
Age	9.23 (7.1–11.3)	10.91 (9.0–12.3)	12.25 (11.3–13.4)	12.50 (12.0–13.0)	0.111	0.384 **
Body weight (Kg)	8 (5.3–8.5)	6.8 (4.7–11.5)	6 (4.9–8.6)	8.5 (6–12.4)	N/A	N/A
Reproductive status [Male:Female (Neutered)]	20:11(10)	15:15(14)	9:3(3)	4:3(3)	N/A	N/A
White blood cells/μL (×10^2^)	9.47 (8.2–10.3)	12.77 (8.1–15.0)	14.00 (9.2–15.0)	11.90 (8.7–15.0)	−0.242 *	0.033
TP (g/dL)	7.93 (7.4–8.2)	7.62 (7.2–8.2)	7.41 (6.7–8.0)	6.60 (5.7–7.0)	0.480 ***	0.091
ALB (g/dL)	3.74 (3.3–3.9)	3.46 (3.3–3.9)	3.69 (3.3–3.9)	3.73 (3.3–4.9)	0.431 ***	0.059
CREA (mg/dL)	0.83 (0.6–0.9)	0.87 (0.6–1.0)	1.08 (0.6–1.1)	1.52 (0.7–1.4)	−0.127	0.095
Heart rate (bpm)	126.12 (108.0–140.0)	127.30 (116.0–144.0)	150.86 (120.0–175.0)	162.00 (140.0–160.0)	−0.082	0.105
Systolic blood pressure (mm Hg)	144.13 (130.0–155.0)	149.7 (128.8–173.0)	134.0 (127.5–171.0)	150.0 (140.0–195.0)	−0.033	−0.148
LVIDdn	1.77 (1.5–1.9)	2.07 (1.7–2.2)	2.19 (2.0–2.3)	2.04 (1.7–2.5)	−0.257 *	0.324 *
LVIDsn	1.00 (0.8–1.1)	1.17 (0.9–1.4)	1.10 (0.9–1.2)	1.08 (0.7–1.5)	−0.283 *	0.181
LA/Ao	1.22 (1.1–1.3)	1.46 (1.4–1.6)	2.08 (1.8–2.5)	1.97 (1.8–2.2)	0.015	0.582 ***
FS%	41.98 (37.2–46.8)	43.95 (38.3–49.8)	48.25 (46.5–53.0)	45.14 (37.5–52.0)	−0.008	0.202
MV E (m/s)	0.84 (0.6–1.0)	1.00 (0.8–1.1)	1.15 (0.8–1.4)	1.14 (0.8–1.4)	0.000	0.089
MV E/A ratio	1.27 (1.1–1.4)	1.51 (1.1–1.9)	1.77 (1.4–1.9)	1.68 (1.3–1.5)	−0.277 *	0.074

* *p* < 0.05, ** *p* < 0.01, *** *p* < 0.001.

**Table 2 vetsci-10-00033-t002:** Descriptive statistics (mean ± standard deviation) with the results of the corresponding testing (ANOVA or *t*-test) for different factors affecting the concentration of PON-1. (PON-1: Paraoxonase-1; MVD: Mitral Valve Disease).

Factor	PON-1	*p*-Value
Clinical Stage of MVD		0.149
B1 (*n* = 31)	4.30 ± 0.96	
B2 (*n* = 30)	3.82 ± 0.88	
C (*n* = 12)	4.13 ± 0.71	
D (*n* = 7)	3.72 ± 0.78	
Gender		0.619
Male (*n* = 48)	4.10 ± 0.90	
Female (*n* = 32)	4.00 ± 0.90	
Coexisting conditions		0.659
Yes (*n* = 21)	4.31 ± 0.89	
No (*n* = 59)	4.21 ± 0.79	

**Table 3 vetsci-10-00033-t003:** Descriptive statistics (median and interquartile range) with the results of the corresponding testing (Mann–Whitney) for different factors affecting the concentration of NT-proBNPin stage B1 and B2 dogs. (NT-proBNP: N-terminal-prohormone-B-type natriuretic peptide, MVD: Mitral Valve Disease).

Factor	NT-proBNP	*p*-Value
Clinical Stage of MVD		<0.001
B1 (*n* = 27)	232 (106–692)	
B2 (*n* = 23)	842 (571–1473)	
Gender		0.573
Male (*n* = 30)	644 (213–944)	
Female (*n* = 20)	593 (156–992)	
Coexisting conditions		0.011
Yes (*n* = 8)	883 (768–944)	
No (*n* = 42)	232 (95–645)	

## Data Availability

Not applicable.

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
