# Peer review of "Association of Paraoxonase-1 and NT-proBNP with Clinical, Clinico-Pathologic and Echocardiographic Variables in Dogs with Mitral Valve Disease"

_vetsci, 2023, doi:10.3390/vetsci10010033_

Round 1

Reviewer 1 Report

The study design looked at NT-proBNP and Paraoxonase-1 in dogs with mitral valve disease (61 dogs for NT-proBNP and 80 for PON-1). The difference in the sample size/population should be more clearly explained. Values for NT-proBNP and Paraoxonase-1 should be compared with those in control animals. Ideally that would be part of the study design. Results from dogs with MVD may be markedly different and/ or similar to control animals. If this cannot be feasibly added experimentally than a discussion on normal ranges and how compare with these study values should be included. If clinicians want to consider using these values they will need to be informed of reference ranges.

Simple summary should be revised to include study information and noted “various” parameters were significant. It is difficult to create a meaningful and brief summary. The authors are encouraged to revise the current attempt with more attention on the study design and less of background information.

Acronyms are used throughout the summary, abstract and introduction that are not defined (e.g. MVD, NT-proBNP, LVIDdn, PON-1, LA/Ao, E/A, FS%) These and others should be define first before being used to improve clarity and readability.

The sentence (line 66) that describes PON-1 as a “moon-lightening protein” provides little value and should be cut.

The sample population should be clearly defined. From line 89, Dogs with MVD were admitted at… Were these patients with suspected MVD or at that already been determined. If so, how was that determined, and my whom?

 The duration of MVD may have a direct impact on the test parameters. Either a statement should be made on the time from diagnosis to testing/progression or the limitations should be expanded to indicate the disease time-course as an uncontrolled variable.

Current medication may be a compounding variable. Were any of the animals on any current medication or therapeutics at the time of the study. Ideally these should be omitted, if not this should be explored and/or discussed as a confounding factor/limitation.

Animal weight is only relevant in the context of body condition. Please include animal body condition in the paper and analysis.

Please include a table with case information including ranges for age, weight, body condition, breed, repro status and murmur grade at the time of inclusion for each clinical stage group. It is unclear how variable each group was.

Who performed the physical and cardiologic examinations? Were these performed by the same person or several. Was interobserver variability controlled for? Please include in the methods.

The current tables provide data ranges without context on their significance. Please include all significant differences and/or P-values in the tables or provide additional figures with this information.

Omit the statement in line 317 and 318 as this was not consistent with the data. Albumin concentration (for example) did not vary significantly with PON-1 value despite a correlation.

The aim of the study as defined in the discussion (line 301) differs from the introduction (line 85-86) and gives very little attention to NT-proBNP. These should be consistent. If the subject of the paper is oxidative stress that should be included in the title, and perhaps NT-proBNP omitted entirely.

If as the title states that PON-1 and NT-proBNP are compared with echo variables that analysis should be included in table 2. Table 2 only include how these variables changes with stage. Table 1 includes how PON-1 and NT-proBNP differ with stage but not echocardiographic variables.

Please include conclusions related to NT-proBNP in symptomatic vs. asymptomatic MVD cases in the conclusions or omit this data from the paper.

Reference 42, 53 and 54 are not formatted consistently with the other references (e.g. use of capitals). These should be revised.

Author Response

Please see the attachment "Response to Reviewer 1" for a point-to-point response to the reviewer's comments

Reviewer 2 Report

See attachment

Author Response

Please see the attachment "Response to Reviewer 2" for a point-to-point response to the reviewer's comments.

Round 2

Reviewer 2 Report

General comment:

I am mostly pleased with the revised version of the manuscript. However, there is one point which I would like to confirm. 

I still could not find any mentions of the other possible concurrent cardiac diseases. In the response from the authors, it appears that 8/80 dogs had tricuspid valve degenerations, which should be mentioned in the manuscript.

Other wise, good work. 

Author Response

We would like to thank the reviewer for his/her comments.

Reviewer 2 comment

I still could not find any mentions of the other possible concurrent cardiac diseases. In the response from the authors, it appears that 8/80 dogs had tricuspid valve degenerations, which should be mentioned in the manuscript.

Authors' response

A sentence has been added in the Materials and methods section (first bullet point) that reads: “..Concurrent tricuspid valve degeneration was not a criterion for exclusion, since the disease progress is the same…”

At the same time, we included a sentence in the results section (fourth paragraph) that reads: “..Tricuspid valve degeneration that resulted in moderate to severe regurgitant volume was evident in 8/80 (10%) dogs…”
